# Phytoplankton Community Structure and Its Relationship with Environmental Factors in Nanhai Lake

**Donghui Gong** [1,2,*], **Ziqing Guo** [1], **Wenxue Wei** [1], **Jie Bi** [1], **Zhizhong Wang** [3] **and Xiang Ji** [4]

1   School of Life Science and Technology, Inner Mongolia University of Science & Technology, Baotou 014010, China
2   Inner Mongolia Key Laboratory for Biomass-Energy Conversion, Baotou 014010, China
3   Department of Biotechnology, Ordos Vocational College of Eco-Environment, Ordos 017010, China
4   Inner Mongolia Agricultural University, Huhot 010019, China
*   Correspondence: gongdh1976@163.com

**Abstract:** In order to determine the characteristics of phytoplankton community structure in Nanhai Lake in Baotou City and its relationship with environmental factors, water and phytoplankton samples were collected and composition and biomass were investigated at six sites in the spring, summer, and autumn of 2021. This article used correlation analysis and redundancy analysis (RDA) combined with the community turnover index (BC) to analyze the phytoplankton functional groups. The results showed that 7 phyla and 68 genera of phytoplankton were identified in the water body, of which Chlorophyta, Bacillariophyta, Cyanobacteria, Chrysophyta, Euglenophyta, Cryptophyta, and Pyrrophyta accounted for 34%, 32%, 16%, 6%, 4%, 4%, and 3%, respectively. The phytoplankton in the water body was classified into 23 functional groups, of which MP and D functional groups were the long−term dominant functional groups, indicating that the habitat is a turbid water body. The ecological state index ($Q$) value ranged from 1.94 to 3.13, with an average value of 2.74. The comprehensive nutritional index ($TSIM(\sum)$) was between 49.32 and 52.11, with an average value of 51.72, indicating that Nanhai Lake was in a mesotrophic state. Correlation analysis and redundancy analysis (RDA) showed that multiple nutrients, transparency (SD), chemical oxygen demand (COD), water temperature (WT), and Chlorophyll a (Chl−a) were the main environmental factors affecting the biomass of dominant functional groups in the water body. The study showed the characteristics of the functional groups of algae in a precious urban lake in arid and semi−arid areas of China and their relationship with environmental factors (physical and chemical indicators, anions and cation ions, and heavy metal ions), and provided a scientific basis for its water quality evaluation.

**Keywords:** phytoplankton; Nanhai Lake; functional groups; community structure; environmental factors





## 1. Introduction

Phytoplankton are a class of tiny plants that grow suspended in water bodies [1]. In aquatic ecosystems, they are primary producers, at the bottom of the food chain, who can use photosynthesis to synthesize organic matter, and are therefore also the basis for energy flow and material circulation [2,3]. Their characteristics of being small individuals, occurring in large numbers, and having short life cycles make them particularly sensitive to environmental changes, which affect the composition, structure, and existing biomass of phytoplankton [4,5].

At present, many studies have been conducted on the structural changes in phytoplankton communities. Most studies reflect the ecological and environmental status of water bodies through conventional methods such as the biomass and diversity index of phytoplankton, which have certain limitations. Therefore, in 2002, on the basis of traditional plant taxonomy, Reynolds [6] first divided the dominant combination of phytoplankton species with the same ecological niche into 31 different functional groups. This was later

expanded, and scholars have now summarized a total of 39 different phytoplankton functional groups. The methods of classifying functional groups are mainly divided into three types, namely, FG (functional groups), MFG (morpho−functional groups), and MBFG (morpho−based functional groups) [6,7]. The FG classification method is used to classify algae in Nanhai Lake. The division of functional groups can not only simplify the study of phytoplankton communities, but also the study of aquatic ecosystems from the perspective of species function, and accurately reflect the relationship between phytoplankton and environmental factors [8].

Nanhai Lake is located in Baotou City, Inner Mongolia Autonomous Region, China; it is supplemented by the Yellow River, and is a precious urban lake in the arid and semi−arid areas of central and western China, having made great contributions to the beautification of the city and the improvement of the ecological environment. In recent years, it has been affected by population growth, the reduction in the Yellow River's water replenishment, increased urban pollution, and the development of industrial and agricultural production. The water quality of the Nanhai wetlands has dropped significantly, various aquatic plants have grown and multiplied in large quantities, the problem of eutrophication has become increasingly serious, and the ecosystem of the lake has been greatly threatened [9]. The wetland area is shrinking, the transparency of water bodies and aquatic biodiversity are decreasing, and environmental pollution is being aggravated, resulting in the increasingly prominent degradation of wetland ecological service functions [10]. Jiang et al. studied the temporal and spatial dynamic characteristics of phytoplankton and their influencing factors in the non−frozen period of Nanhai Lake, and found that the Chlorophyta was closely related to total nitrogen, Cyanobacteria was mainly affected by total phosphorus, and Bacillariophyta was greatly affected by pH [11]. Yang et al. used the ecological niche method and the gray association analysis method to study the dominant species of phytoplankton during the freeze–thaw period in Nanhai Lake, and found that the combination of the two methods can adequately describe the community ecological characteristics of the dominant species of phytoplankton, and that *Scenedesmus quadricauda* is the most stable of the dominant species across the whole process [12]. However, there is currently little research [13] on the main environmental drivers of structural changes in phytoplankton functional groups in the water body. Therefore, for the first time, the content of physicochemical factors, salt ions, and heavy metal ions in the water body was systematically determined. The water quality of the study area was evaluated by the ecological state index (*Q*) and the comprehensive nutritional index (*TSIM*). Using redundancy analysis (RDA) and correlation analysis, combined with the stability of dominant communities, the ecological characteristics of phytoplankton functional groups and their relationship with environmental factors were discussed. The main goal of the study was not only to provide scientific data for pollution of the water body, but also to provide a basic scientific basis for its water quality evaluation.

## 2. Materials and Methods

### 2.1. Sample Settings

Nanhai Lake (40°30′8″~40°33′32″ N, 109°59′26″~110°2′26″ E) is located in the arid and semi−arid grassland area and belongs to the temperate arid and semi−arid continental monsoon climate zone. The lake area is $333 \times 10^4$ m$^2$, the maximum water depth reaches 3 m, and the lake is surrounded by dense reeds. In July, the temperature is high, the solar radiation is strong, the rainfall is large, and the water level and temperature of the lake rise. After entering September, the rainfall decreases and the temperature gradually decreases. Six sites (N1, N2, N3, N4, N5, and N6) were selected in the sampling area, and water samples were collected at these six sampling sites every two months (in the first week of May, July, and September) in 2021, measuring three months in parallel with monthly data taking the average of six sampling sites. The sampling point layout is shown in Figure 1.

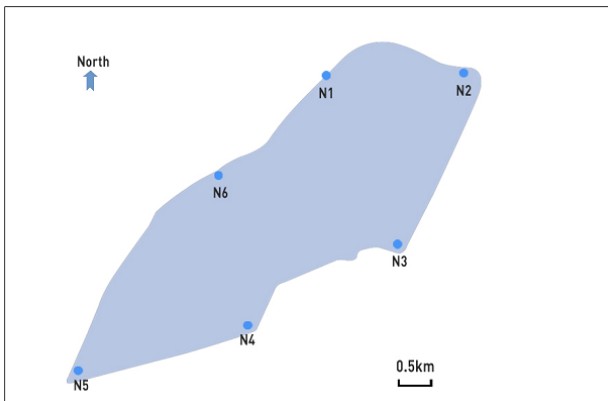

**Figure 1.** Distribution of sampling sites (N1–N6) in Nanhai Lake.

### 2.2. Phytoplankton Collection Methods and Identification

Both qualitative and quantitative methods were used to analyze the phytoplankton distribution. Qualitative samples were taken on the surface of the water body with a No. 25 plankton net (net length: 50 cm, net ring diameter: 20 cm, aperture: 0.064 mm) for phytoplankton identification, which was dragged back and forth on the water surface in a "∞" shape for 1–3 min, before being lifted and the excess water filtered out. The samples were poured into sampling bottles, approximately 100 mL of concentrated water samples was collected, and then 4% formaldehyde solution was immediately added for fixation. Quantitative samples (the mixture of different layers) were poured into 1 L plastic sampling bottles before adding 4% formaldehyde solution dropwise, shaking well, and fixing the samples on the spot. The samples were brought back to the laboratory at 4 °C for storage and were counted under the microscope using pipettes of 0.1 mL water samples. Phytoplankton counts were performed using a 0.1 mL phytoplankton counting chamber (20 mm × 20 mm, JHH 0.1 mL, Jinhuihuang Scientific and Technical Corporation, China) with an Olympus CX23 microscope at 400× magnification. The second, fifth, and eighth rows of the counting chamber were scanned yielding a total of 30 squares. All phytoplankton cells in each square were counted and identified. Three replicates were performed per sample, taking *The Freshwater Algae of China: System, Ecology and Classification* [14] and *Freshwater Algae of North America: Ecology and Classification* [15] as the basis for the identification of phytoplankton species.

### 2.3. Determination of Physical and Chemical Indicators

A portable tester (Hanna, HI98194, Italy) was used on site to determine water temperature (WT) (°C), transparency (SD) (m), total solid solubility (TDS) (mg/L), dissolved oxygen (DO) (mg/L), conductivity (EC) (μs/cm), and pH. Taking a 1 L water sample to determine the physical and chemical indicators, the main method referred to the fourth edition of "*Water and Wastewater Monitoring and Analysis Methods*" [16], as shown in Table 1.

**Table 1.** Indicators and methods for the detection of Nanhai Lake samples.

| Serial Number | Index | Analytical Method |
|---|---|---|
| 1 | Ammonia Nitrogen ($NH_3-N$) (mg/L) | Nessler's Reagent Spectrophotometry |
| 2 | Total Phosphorus (TP) (mg/L) | Ammonium Molybdate Spectrophotometric Method |
| 3 | Chlorophyll a (Chl−a) (μg/L) | Acetone Extraction Method |
| 4 | Chemical Oxygen Demand (COD) (mg/L) | Rapid Digestion Method |
| 5 | Biochemical Oxygen Demand ($BOD_5$) (mg/L) | Dilution and Seeding Method |
| 6 | $Na^+$, $Mg^{2+}$, $K^+$, $Ca^{2+}$ (mg/L) | ICP (Inductively Coupled Plasma Emission Spectrometer) (Agilent, 725−ES, USA) |
| 7 | $Cl^-$, $SO_4^{2-}$, $CO_3^{2-}$, $HCO_3^-$ (mg/L) | Titrimetric Method |

**Table 1.** *Cont.*

| Serial Number | Index | Analytical Method |
|:---:|:---:|:---:|
| 8 | $NO_3^-$ (mg/L) | Spectrophotometric Method (Model−752) |
| 9 | Heavy metal ion (mg/L) | ICP (Inductively Coupled Plasma Emission Spectrometer) (Agilent, 725−ES, USA) |

*2.4. Data Analysis*

Key data collation and analysis were completed in Microsoft Excel 2010, and ORGIN 2019 (OriginLab, USA) was used to draw statistical charts. Analysis of variance (ANOVA) was used with SPSS software (IBM, USA) to make multiple comparisons of intragroup differences and analyze the relationship between the biomass of the dominant functional group and the environmental factors. Combined with scatter plots, the distribution range of the biomass of the dominant functional group was predicted under certain environmental conditions. Canoco5 (Microcomputer Power, USA) was used for detrend correspondence analysis (DCA); when the length of the sorting axis was less than three, the relationship between dominant functional group and environmental factors was studied by redundancy analysis (RDA). The Monte Carlo displacement test was used to automatically filter the environment variables during the analysis to obtain a sorting plot between the phytoplankton functional group and the environmental factors [17].

2.4.1. Functional Group Classification and Biomass (*B*)

The phytoplankton functional group classification referenced the phytoplankton functional group classification method proposed by Reynolds et al. [6]. The formula for calculating the biomass of phytoplankton is as follows [18]:

$$B = (N/T) \times M \times K \times 0.0001$$

*N*: number of algae; *T*: total number of algae; *M*: total phytoplankton of 1 L; *K*: average wet weight of phytoplankton cells.

2.4.2. Bray–Curtis Heterogeneity Index (*BC*)

The Bray–Curtis Heterogeneity Index (*BC*) is used to measure community stability, with a value range of [0, 1]. The higher the *BC* value, the stronger the species turnover and the weaker the stability. The calculation formula is as follows [19]:

$$BC = \sum_{i=1}^{n} |y_{i1} - y_{i2}| \, / \sum_{i=1}^{n} \left| y_{i1} + y_{i2} \right|$$

In the equation, *BC* represents the community turnover index; $y_{i1}$ is the first measured biomass (mg/L) for species *i* in the community; $y_{i2}$ is the second measured biomass (mg/L) for species *i*; and *n* is the total number of species in the community identified by the two measurements.

2.4.3. Comprehensive Nutrient Status Index (*TSIM*)

The degree of eutrophication of the water body was analyzed by the comprehensive nutrient status index (*TSIM*) method [20]. The evaluation index of this method mainly selects Chl−a, TP, and SD in the lake. The calculation formula is as follows:

$$TSIM\ (Chl-a) = 10 \times (2.46 + lnChl-a\ /ln2.5)$$

$$TSIM\ (SD) = 10 \times (2.46 + (3.69 - 1.53)\ lnSD\ /ln2.5)$$

$$TSIM\ (TP) = 10 \times (2.46 + (6.71 + 1.15lnTP\ /ln2.5)$$

$$Total\ TSI = (TSIM\ (Chl-a) + TSIM\ (SD) + TSIM\ (TP))\ /3$$

The evaluation criteria are as follows: *TSI* < 37 is a poorly nutritious type; 38 < *TSI* ≤ 53 is a moderately nutritious type; and *TSI* > 53 is rich in nutrients. In general, the higher the index, the more nutritious the water body is.

### 2.4.4. Ecological State Index (*Q*)

The *Q* index is used to evaluate and analyze the water quality; the calculation formula is as follows [21]:

$$Q = \sum PiFi$$

In the equation, *n* is the number of functional groups; *Pi* is the ratio of the biomass to the total biomass of the *ith* functional group; the number of factors *Fi* is the assignment of the *ith* functional group; and the *Q* index is between 0 and 5 [22]. The evaluation criteria are 0~1 poor, 1~2 tolerable 2~3 medium, 3~4 good, and 4~5 excellent [23].

## 3. Results

### 3.1. Division of Phytoplankton Species and Functional Groups

From May to September 2021, a total of 7 phyla and 68 genera of phytoplankton were identified in the water body. Among them, Chlorophyta represented 23 genera (34%), Bacillariophyta represented 22 genera (32%), and Cyanobacteria represented 11 genera (16%). Chlorophyta, Bacillariophyta, and Cyanobacteria were the main groups of phytoplankton species (82%). The rest of the algae were represented by relatively few species; Chrysophyta represented four genera (6%), there were three genera (4%) each of Euglenophyta and Cryptophyta, and Pyrrophyta was represented by two genera (3%). *Synedra* sp. and *Spirulina* sp. were the dominant algae species in May and September, while no dominant species appeared in July.

Reynolds et al. [6] proposed a method for the functional group classification of planktonic algae, and the results are displayed in Table 2. A total of 23 functional groups were identified: functional groups Y, B, J, X1, Lo, D, M, X2, X3, Wo, S1, A, E, TB, MP, P, G, H1, N, F, TC, S2, and W1. The frequency of occurrence for each functional group is shown in Figure 2. Functional groups D, F, Lo, J, M, MP, P, X1, and X3 appeared with a frequency of more than 90%, making them the dominant functional groups of spring, summer, and autumn during the study. Functional groups B, N, S2, TC, and W1 occur with a frequency between 50% and 90%, making them common functional groups. Functional groups Wo, S1, Y, E, H1, G, and TB occurred between 20% and 50% of the time, and only occurred in suitable habitats. The frequency of functional groups A and X2 was not more than 20%, which was low, making them occasional or rare functional groups.

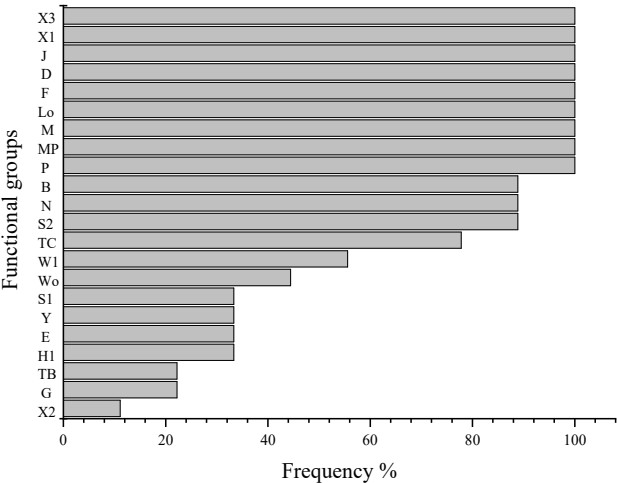

**Figure 2.** Frequency of phytoplankton functional groups in Nanhai Lake.

**Table 2.** Functional group composition of phytoplankton in Nanhai Lake.

| Functional Group | Typical Representatives | Habitat | May | Time July | Sept. |
|---|---|---|---|---|---|
| A | *Rhizosolenia* sp. | Clear, often well−mixed, base−poor lakes | | | * |
| B | *Cyclotella* sp. | Vertically mixed, mesotrophic small–medium lakes | * | * | * |
| D | *Synedra* sp., *Nitzschia* sp. | Shallow, enriched turbid waters, including rivers | * | * | * |
| E | *Dinobryon* sp. | Usually small, oligotrophic, base−poor lakes or heterotrophic ponds | | | * |
| F | *Kirchneriella* sp., *Oocystis* sp., *Selenastrum* sp. | Clear epilimnia | * | * | * |
| G | *Eudorina elegans* sp. | Short, nutrient−rich water columns | | | * |
| H1 | *Anabaena* sp., *Pseudanabaena* sp., *Aphanizomenon* sp. | Dinitrogen−fixing Nostocaleans | | | * |
| J | *Scenedesmus* sp., *Tetraedron* sp., *Coelastrum* sp., *Crucigenia* sp., *Pediastrum* sp. | Shallow, enriched lakes, ponds, and rivers | * | * | * |
| Lo | *Chroococcus* sp., *Merismopedia* sp., *Peridinium* sp., *Gymnodinium* sp., *Pinnularia* sp. | Summer epilimnia in mesotrophic lakes | * | * | * |
| M | *Microcystis* sp. | Ideally mixed layers of small, eutrophic, low−latitude lakes | * | * | * |
| MP | *Navicula* sp., *Cymbella* sp., *Achnanthes* sp. | Frequent agitation, turbidity, shallow water | * | * | * |
| N | *Cosmarium* sp., *Staurastrum* sp. *Tabellaria* sp. | Mesotrophic epilimnia | * | * | * |
| P | *Fragilaria* sp., *Melosira* sp. *Closterium* sp. | Eutrophic epilimnia | * | * | * |
| TB | *Gomphonema* sp. | Rapid−flowing water | | | * |
| TC | *Oscillatoria* sp. | Nutrient−rich static water bodies, or slow−flowing rivers with algae outbreaks | * | * | * |
| S1 | *Lyngbya* sp. | Turbid mixed layers | * | | * |
| S2 | *Spirulina* sp., *Synedra* sp. | Shallow, turbid mixed layers | * | * | * |
| Wo | *Chlamydomonas* spp. | Rivers and ponds rich in organic matter or humus | * | | * |
| W1 | *Euglena* sp., *Phacus* sp. | Small organic ponds | | | * |
| X1 | *Ankistrodesmus* sp. | Shallow mixed layers in enriched conditions | * | * | * |
| X2 | *Chroomonas* sp. | Shallow, clear mixed layers in meso−eutrophic lakes | | | * |

**Table 2.** *Cont.*

| Functional Group | Typical Representatives | Habitat | Time | | |
|---|---|---|---|---|---|
| | | | May | July | Sept. |
| X3 | *Chlorella* sp., *Schroederia* sp., *Ochromonas* sp. | Shallow, clear, mixed layers | * | * | * |
| Y | *Cryptomonas* sp., *Gymnodinium* sp. | Usually, small, enriched lakes | | * | * |

Note: * indicates that the functional group appears in this month.

Functional groups with a relative biomass greater than 5% for at least one sampling point were defined as dominant functional groups. This can be seen from the relative biomass and total biomass of the functional groups of phytoplankton in the lake (Figure 3). The phytoplankton biomass ranged from 0.15 to 1.01 mg/L, being 0.15 to 0.28 mg/L in May, 0.16 to 0.27 mg/L in July, and fluctuating between 0.57 and 1.01 mg/L in September. The total biomass in September was significantly higher than in May and July ($p < 0.01$). In May, Nanhai Lake displayed functional groups MP, M, J, and D as its dominant functional groups, of which the MP functional group accounted for the largest proportion, being 47%. In July, functional groups D, F, J, Lo, M, MP, P, TC, and S2 were the dominant functional groups, and there were more advantageous functional groups that were also more evenly distributed. Functional groups D, MP, Lo, and S2 were the dominant functional groups in September, of which functional groups D and S2 accounted for the highest proportions of 61% and 62%, respectively. Functional groups D and MP were advantageous functional groups which appeared steadily over 3 months.

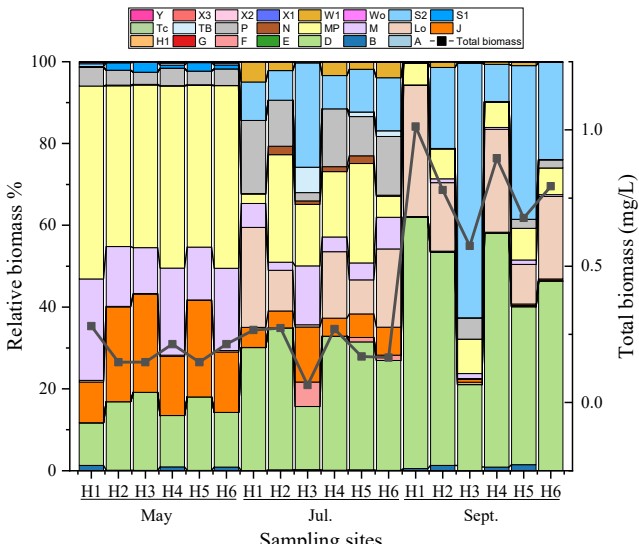

**Figure 3.** Seasonal variation of relative biomass and total biomass among functional groups of phytoplankton at six sampling sites (N1–N6) in Nanhai Lake.

### 3.2. Water Quality Assessment Based on Phytoplankton Communities

3.2.1. Community Stability

Through the analysis of the community stability of nine dominant functional groups from May to September (Table 3), it was found that there were turnovers and fluctuations in the stability of the dominant functional groups in different months. The community stability fluctuation range of functional groups D and MP was small, and the community stability of each month improved. The BC values were below 0.5, indicating that the D and MP functional groups maintained a relatively stable dominant position from May to September. The BC value of the J functional group varied the most. Functional group J

had a low BC value from May to July and a high BC value from July to September, so the dominance of functional group J in the spring was stable. In contrast, functional groups Lo and S2 had higher BC values from May to July and lower BC values from July to September, so the dominance of functional groups Lo and S2 was stable in the summer. The BC values of the P, F, M, and TB functional groups were larger from May to September, indicating that the degree of turnover for each functional group during this period was relatively unstable.

**Table 3.** The community turnover values for the dominant phytoplankton functional groups in Nanhai Lake.

| Community Turnover Value (BC) | D | MP | J | S2 | Lo | P | F | M | TB |
|---|---|---|---|---|---|---|---|---|---|
| May−July | 0.15 | 0.36 | 0.15 | 1.00 | 1.00 | 1.00 | 1.00 | 0.96 | 1.00 |
| July−Sept. | 0.22 | 0.43 | 1.00 | 0.54 | 0.39 | 1.00 | 1.00 | 1.00 | 1.00 |

### 3.2.2. Ecological Status Index of Phytoplankton Functional Groups

Water quality was evaluated with the use of the ecological index ($Q$) values for phytoplankton functional groups (Figure 4). From May to September, the $Q$ value was between 1.94 and 3.13, with an average of 2.74, and the overall water quality of the study area was at a medium level in terms of time. In July, the $Q$ value was relatively low, being at the tolerable level, and the water quality was poor. The results of the evaluation of the nutritional status of the water body by the *TSIM* index showed that the monthly *TSIM* values during the survey period ranged from 49.32 to 52.11, and the average value was 51.72. According to the evaluation criterion of $38 < TSI \leq 53$ for the moderately nutritious type, the water quality was in the transition stage of the moderately rich nutrient state, which was consistent with the $Q$ index results, and the overall trend was that the larger the $Q$ value, the smaller the *TSIM* value.

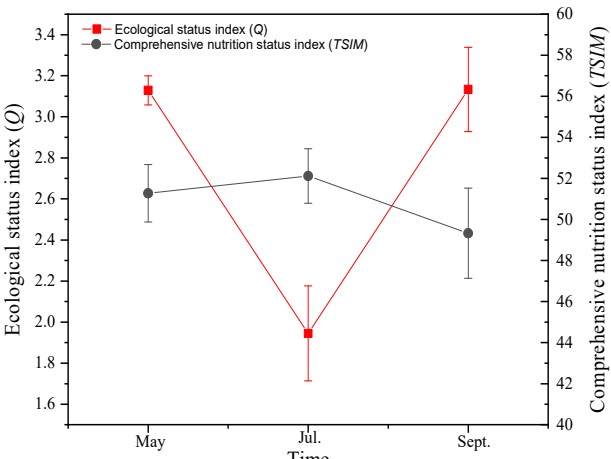

**Figure 4.** The temporal distribution of ecological status index ($Q$) and comprehensive nutrition status index (*TSIM*) ($\pm$S.E).

### 3.2.3. Environmental Factors

The results of the Nanhai Lake quality test are shown in Figure 5. The pH of the water body was greater than 9, and the lake water was weakly alkaline. The water temperature ranged from 18.49 to 31.08 °C; it began to rise in May, reached a maximum (31.08 °C) in July, and gradually decreased in September, with an average of 23.60 °C. TDS and EC exhibited the same trend as WT, with TDS ranging from 1047 to 1105 mg/L, with an average of 1078 mg/L, and EC ranging from 2096 to 2210 μs/cm, with an average of 2156 μs/cm; the TDS and EC values in May differ significantly from those in July and September ($p < 0.05$). SD, BOD$_5$, NH$_3$−N, and TP exhibit the same trend, with the highest values in May, and the lowest values in September, with the range of SD changes being 0.31 ~ 0.70 m, and

the average value 0.54 m. $BOD_5$ varied from 1.13 to 23.20 mg/L, with a mean value of 7.11 mg/L. $NH_3-N$ and TP ranged from 3.82 to 9.42 mg/L and 0.21 to 0.66 mg/L, respectively, with significant differences ($p < 0.05$) over three months, and had means of 6.80 mg/L and 0.37 mg/L, respectively. COD and Chl−a values showed a gradual upward trend from May to September. COD ranged from 3.58 to 12.23 mg/L, with an average value of 6.63 mg/L, and the COD value in September was significantly different from the COD value in May and July ($p < 0.05$). Chl−a varied from 24.90 to 62.11 μg/L, with a mean value of 47.06 μg/L, and the Chl−a value in May was significantly different from the Chl−a value in July and September ($p < 0.05$). The average of DO was 8.16 mg/L, with a maximum of 10.12 mg/L in May and a minimum of 5.25 mg/L in July.

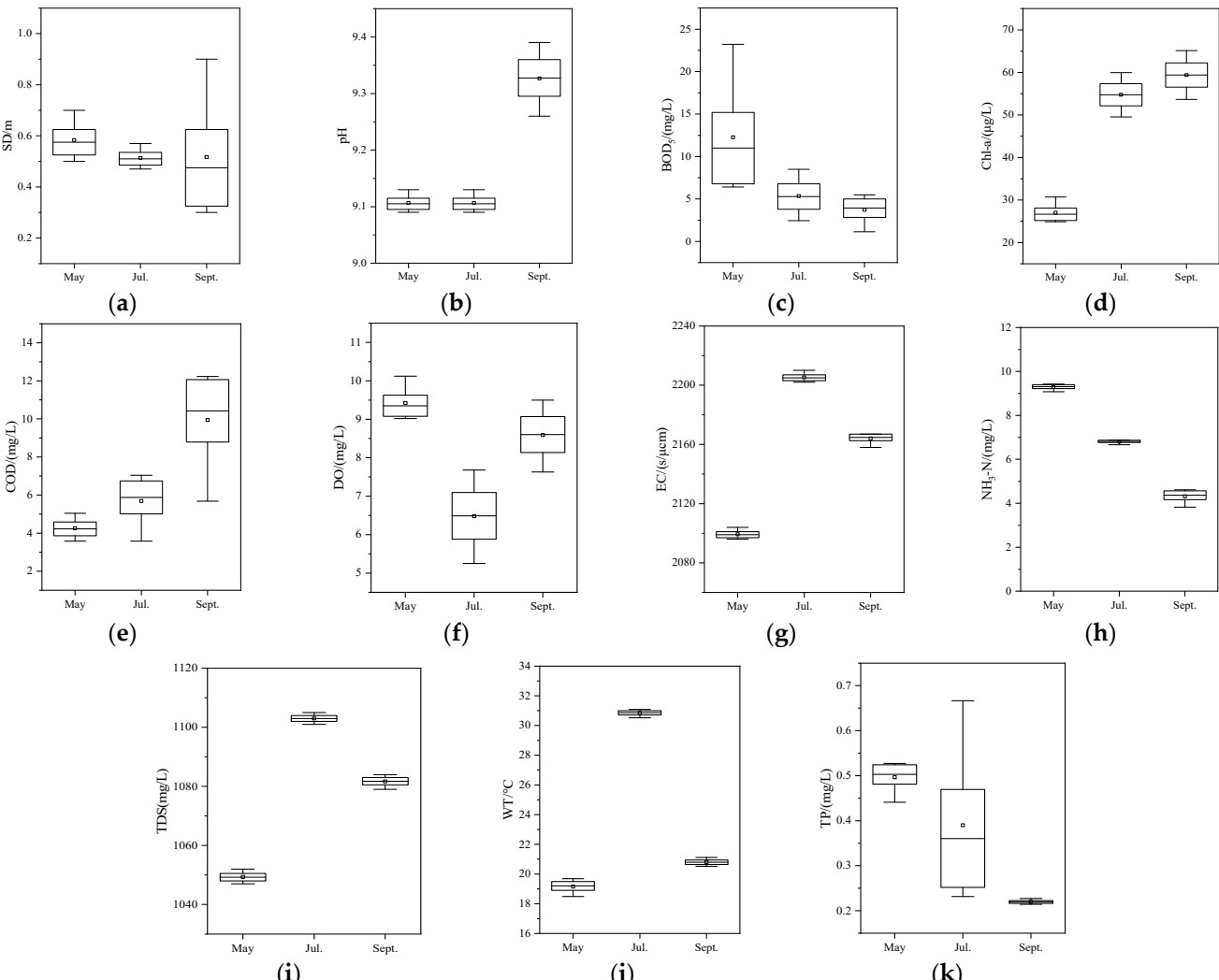

**Figure 5.** Physical and chemical indicators of water quality at different sampling sites (±S.E). Notes: (**a**): SD. (**b**): pH. (**c**): $BOD_5$. (**d**): Chl−a. (**e**): COD. (**f**): DO. (**g**): EC. (**h**): $NH_3-N$. (**i**): TDS. (**j**): WT. (**k**): TP.

We refer to the "GB Surface Water Environmental Quality Standard" (3838−2002) for our evaluation. The $BOD_5$ maximum in May was of Class V water standard (10 mg/L), and in July and September it was of Class III surface water standard (4 mg/L). $NH_3-N$ exceeded the Class V surface water range (2 mg/L) throughout the year and exceeded the standard more severely on occasion. In May, July, and September, the TP values reached the standard limits of Class V (0.4 mg/L), Class IV (0.3 mg/L), and Class III (0.2 mg/L), respectively, and there was a trend of gradual improvement.

From the perspective of anion content (Figure 6), the content of cations in the water from May to September ranged from 441.16 to 543.29 mg/L, showing the characteristics of $Na^+ > Mg^{2+} > Ca^{2+} > K^+$. The anions ranged from 1223.24 to 1349.53 mg/L in the order $Cl^- > SO_4^{2-} > HCO_3^- > CO_3^{2-} > NO_3^- > F^-$. The cation content was low in July (441.16 mg/L), and the difference was not significant in May (543.29 mg/L) and September (518.24 mg/L) ($p > 0.05$). The anion content was the lowest in September (1223.24 mg/L), and the difference was not significant in May (1347.80 mg/L) and July (1349.53 mg/L) ($p > 0.05$). The heavy metal content is shown in Table 4, and the difference in heavy metal content from May to September was not significant ($p > 0.05$), with the Sr content being higher, followed by Li and Ba. According to the "GB Surface Water Environmental Quality Standard" (3838−2002), it was found that the $SO_4^{2-}$ and $Cl^-$ content seriously exceeded the standard limits (250 mg/L), and $F^-$ and $NO_3^-$ also exceeded the standard limits (the standard limit of $F^-$ was 1.5 mg/L, and of $NO_3^-$ was 10 mg/L), but the excess was not significant ($p > 0.05$) in the three seasons studied. The $Na^+$ content far exceeded the range of Class III surface water ($\geq$200 mg/L), and none of the heavy metals exceeded the standard in the water body.

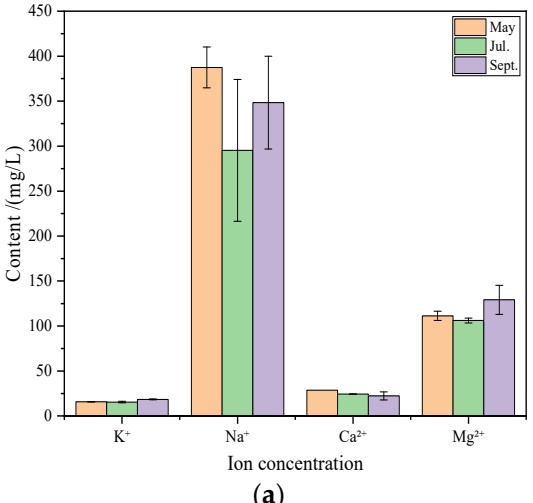

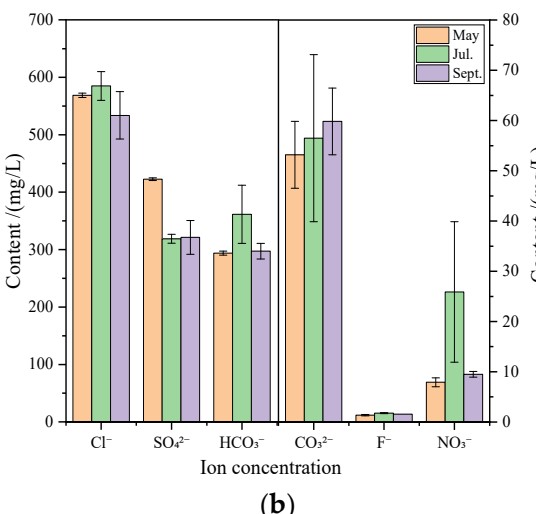

(a)  (b)

**Figure 6.** The variations of anion and cation concentration in Nanhai Lake ($\pm$S.E). Notes: (**a**): anion content in May, July, and September; (**b**): cation content in May, July, and September.

**Table 4.** The variations of heavy metal concentration in Nanhai Lake (mean $\pm$ S.E).

| Heavy Metal Concentration | May | Jul. | Sept. |
|---|---|---|---|
| Al (mg/L) | 0.0050 ± 0.0035 | 0.0163 ± 0.0009 | 0.0225 ± 0.0053 |
| Ba (mg/L) | 0.0850 ± 0.0035 | 0.1010 ± 0.0014 | 0.0765 ± 0.0025 |
| Sr (mg/L) | 0.6050 ± 0.0177 | 0.3252 ± 0.0037 | 0.2180 ± 0.0035 |
| Ag (mg/L) | 0.0000 ± 0.0000 | 0.0000 ± 0.0000 | 0.0000 ± 0.0000 |
| As (mg/L) | 0.0038 ± 0.0003 | 0.0115 ± 0.0004 | 0.0100 ± 0.0007 |
| Cd (mg/L) | 0.0000 ± 0.0000 | 0.0000 ± 0.0000 | 0.0000 ± 0.0000 |
| Cr (mg/L) | 0.0001 ± 0.0000 | 0.0000 ± 0.0000 | 0.0000 ± 0.0000 |
| Fe (mg/L) | 0.0023 ± 0.0003 | 0.0275 ± 0.0018 | 0.0050 ± 0.0014 |
| Mn (mg/L) | 0.0003 ± 0.0000 | 0.0105 ± 0.0011 | 0.0050 ± 0.0014 |
| Pb (mg/L) | 0.0001 ± 0.0000 | 0.0000 ± 0.0000 | 0.0000 ± 0.0000 |
| Co (mg/L) | 0.0002 ± 0.0000 | 0.0000 ± 0.0000 | 0.0000 ± 0.0000 |
| Cu (mg/L) | 0.0004 ± 0.0000 | 0.0000 ± 0.0000 | 0.0000 ± 0.0000 |
| Li (mg/L) | 0.0519 ± 0.0009 | 0.0651 ± 0.0004 | 0.0485 ± 0.0011 |
| Ni (mg/L) | 0.0012 ± 0.0000 | 0.0010 ± 0.0000 | 0.0010 ± 0.0000 |
| Se (mg/L) | 0.0001 ± 0.0000 | 0.0000 ± 0.0000 | 0.0000 ± 0.0000 |
| Zn (mg/L) | 0.0002 ± 0.0001 | 0.0000 ± 0.0000 | 0.0000 ± 0.0000 |

### 3.3. Relationship between Phytoplankton and Environmental Factors

#### 3.3.1. Relationship between Water Quality Assessment Indicators and Environmental Factors

Pearson correlation analysis was performed between all environmental factors and the *Q* index and *TSIM* index from May to September (Figure 7), and the results showed that the *Q* index had a significant positive correlation with SD ($p < 0.01$), and a very significant ($p < 0.01$) and significant ($p < 0.05$) negative correlation with $Mg^{2+}$ and $HCO_3^-$, respectively. The *TSIM* index was significantly positively correlated with $Ca^{2+}$ ($p < 0.05$), significantly positively correlated with $SO_4^{2-}$ ($p < 0.05$), and very significantly ($p < 0.01$) and significantly negatively correlated with $Cl^-$ and TDS ($p < 0.05$), respectively. Overall, the relationship between the salt ions $Mg^{2+}$, $HCO_3^-$, $Ca^{2+}$, $SO_4^{2-}$, and $Cl^-$ and water quality evaluation indicators was significant ($p < 0.05$) or even very significant ($p < 0.01$), indicating that the change in nutrient salt concentration was closely related to the *Q* index and *TSIM* index.

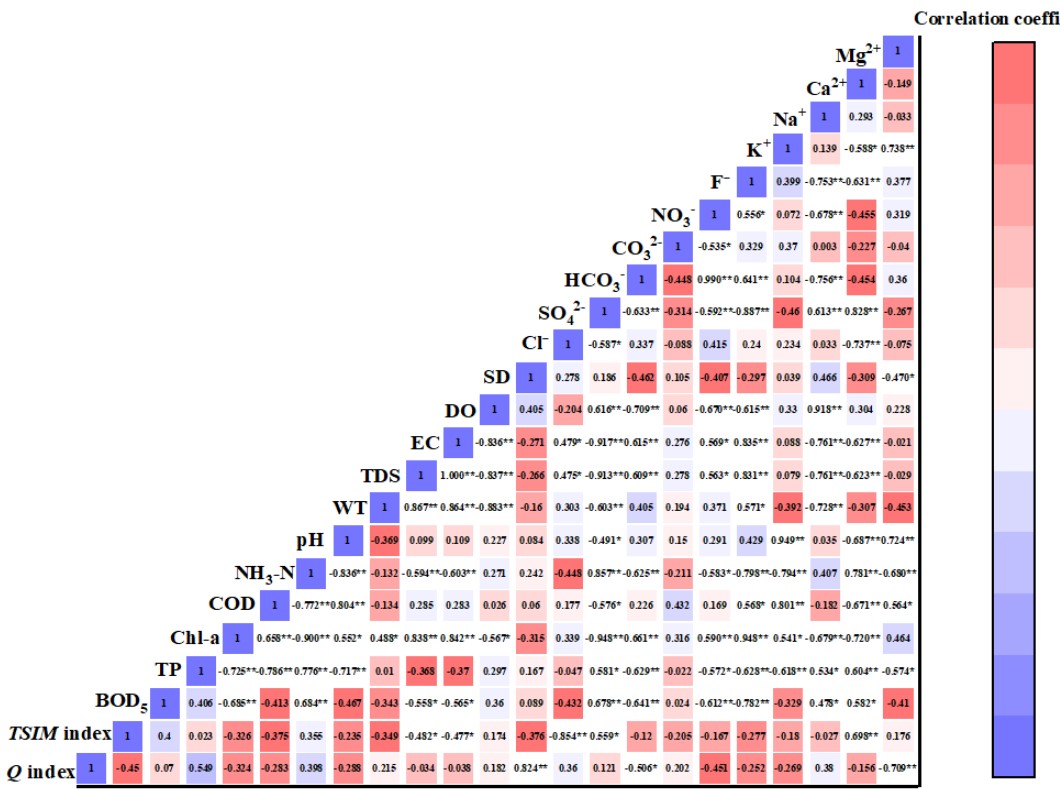

**Figure 7.** Pearson correlation analysis of environmental factors and *Q* index and *TMIS* index in Nanhai Lake. Notes: ** Indicates that the correlation is significant at the 0.01 level (double tails), * indicates that the correlation is significant at the 0.05 level (double tails).

#### 3.3.2. Spearman Correlation Analysis of Dominant Functional Groups and Environmental Factors

Correlation analysis of the dominant functional group biomass and environmental factors in Nanhai Lake was conducted, and the results are displayed in Figure 8. Functional group D was correlated with a variety of environmental factors, which were significantly negatively correlated with TP ($p < 0.05$), very significantly positively correlated with WT and Chl−a ($p < 0.01$), significantly negatively correlated with $SO_4^{2-}$ concentration ($p < 0.05$), and very significantly positively correlated with $NO_3^-$ ($p < 0.01$). Functional group J was significantly negatively correlated with Chl−a ($p < 0.01$), functional group Lo was significantly negatively correlated with TP ($p < 0.01$), $Ca^{2+}$ was significantly negatively correlated with functional group Lo ($p < 0.01$), and functional group S2 and $K^+$ were significantly positively correlated ($p < 0.01$). The correlation between the biomass of the dominant functional groups of phytoplankton and the environmental factors was analyzed

for the water body, and the distribution of the environmental factors that were significantly related to the biomass of the dominant functional groups was predicted. Functional group D appeared in TP concentrations of 0.21 to 0.27 mg/L, WT of 18.49 to 21.12 °C, Chl−a of 54.76 to 59.96 μg/L, $SO_4^{2-}$ of 311.23 to 326.60 mg/L, and $NO_3^-$ of 11.90 to 12.80 mg/L. Functional group J appeared in the Chl−a concentration range of 56.52 to 62.25 μg/L. Functional group Lo appeared in the range of TP concentrations between 0.21 and 0.27 mg/L and $Ca^{2+}$ concentrations of 23.94 to 25.33 mg/L. Functional group S2 concentrations occurred at $K^+$ concentrations between 15.46 and 16.16 mg/L.

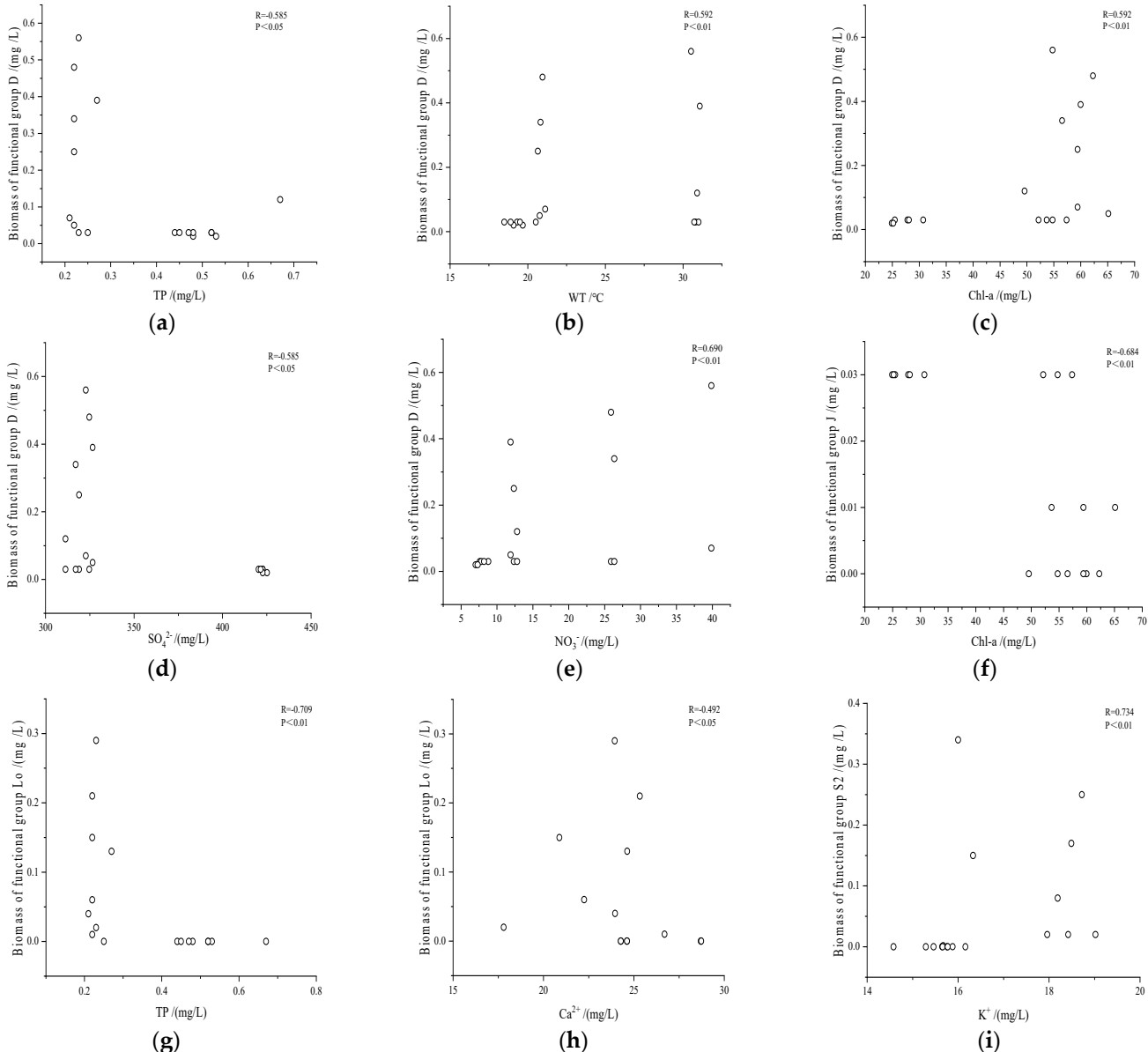

**Figure 8.** Relationship between biomass of dominant functional groups of phytoplankton and some water environmental factors (TP, WT, Chl−a, $SO_4^{2-}$, $NO_3^{2-}$, $Ca^{2+}$, and $K^+$) in Nanhai Lake. Notes: (**a**): relationship between biomass of functional group D and TP; (**b**): relationship between biomass of functional group D and WT; (**c**): relationship between biomass of functional group D and Chl−a; (**d**): relationship between biomass of functional group D and $SO_4^{2-}$; (**e**): relationship between biomass of functional group D and $NO_3^{2-}$; (**f**): relationship between biomass of functional group J and Chl−a; (**g**): relationship between biomass of functional group Lo and TP; (**h**): relationship between biomass of functional group Lo and $Ca^{2+}$; (**i**): relationship between biomass of functional group S2 and $K^+$.

### 3.3.3. Redundancy Analysis of Functional Group Strength and Environmental Factors

In order to explore the effects of environmental factors on the community structure of phytoplankton, first the biomass removal trend (DCA) of the dominant functional group was analyzed. The results showed that the maximum gradient length of the sort axis was less than three, so redundancy analysis (RDA) was used for data selection (Figure 9). The Monte Carlo fitting method was used to test the significance of the factors. The Monte Carlo fitting method was applied to test the significance of the factors, and by screening $Mg^{2+}$ ($p = 0.002$, $F = 32.6$), COD ($p = 0.002$, $F = 12.3$), $NH_3-N$ ($p = 0.002$, $F = 14.4$), $Na^+$ ($p = 0.002$, $F = 9.6$), WT ($p = 0.002$, $F = 16.5$), SD ($p = 0.002$, $F = 12.5$), $K^+$ ($p = 0.002$, $F = 14.6$), and Chl$-$a ($p = 0.02$, $F = 13.5$) were found to be significant explanatory variables, representing 67.1%, 14.8%, 9.3%, 3.8%, 3.0%, 1.1%, 0.9%, and 0.1% of the community variation, respectively, therefore being the key environmental factors affecting phytoplankton community variation. The environmental factors that did not have a significant impact are represented by dotted lines in Figure 9. The RDA analysis showed a certain temporal heterogeneity, with the second quadrant in May, the first and fourth quadrants in September, and the third and fourth quadrants in July. The first and second sort axes feature values of 0.5943 and 0.1825, respectively, explaining 59.43% and 77.67% of the biomass information, respectively. The first sort axis was positively correlated with $Mg^{2+}$, $K^+$, COD, Chl$-$a, and WT, and was negatively correlated with SD, $Na^+$, and $NH_3-N$. The second sort axis was positively correlated with $Mg^{2+}$, $K^+$, $Na^+$, and $NH_3-N$, and was negatively correlated with SD, COD, Chl$-$a, and WT. Functional group MP was significantly negatively correlated with WT, functional groups TC, D, and Lo were significantly positively correlated with $Mg^{2+}$ and $K^+$, functional group S2 was significantly positively correlated with COD, Chl$-$a, and WT, while WT and SD were significantly positively correlated with functional groups P and F, and MP, J, and M were significantly positively correlated with $NH_3-N$ and $Na^+$.

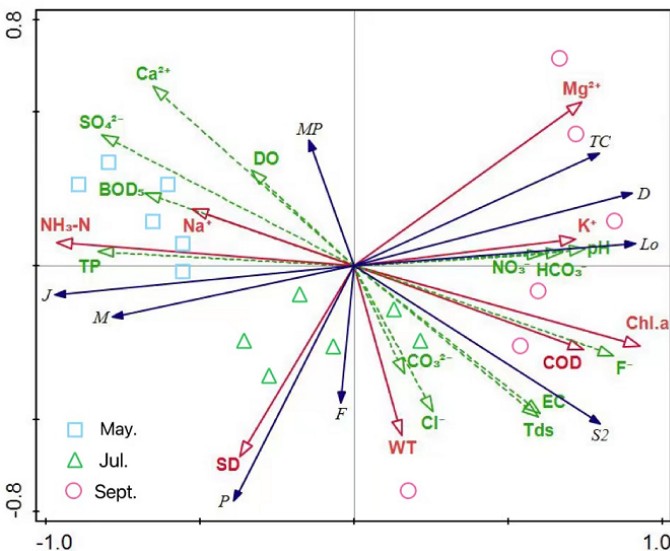

**Figure 9.** RDA$-$analysis$-$based relationship between the main species of phytoplankton and water quality indexes in Nanhai Lake.

## 4. Discussion

### 4.1. Characteristics of the Phytoplankton Community

The phytoplankton community was mainly composed of Chlorophyta, Bacillariophyta, and Cyanobacteria. Chlorophyta was the dominant population, with more species and the largest proportion (34%), which somewhat differed from the test results of phytoplankton in the spring obtained by Jiang Qinghong et al. [24]; this may be related to changes in the water environment between different year. The proportions of Cyanobacteria and Chlorophyta were mainly due to the gradual increase in temperature in Baotou City since May. The nitrogen and phosphorus concentrations were also higher, which is beneficial

for the growth of algae that are tolerant to high temperatures and prefer high nutrient levels [25]. Bacillariophyta are usually more abundant in spring and autumn, and unpolluted water bodies contain mostly Bacillariophyta, while Chlorophyta, Bacillariophyta, and Cyanobacteria are relatively rare. Previous studies have shown that the water body of the Huaxi River has been polluted by industry, urban life, and agriculture [26]. Therefore, some scholars believe that the water pollution of the Huaxi River in spring, summer, and autumn has a greater impact on the composition of phytoplankton, resulting in the community composition being dominated by Chlorophyta [27].

The total biomass of phytoplankton in the study area changed regularly at different times; the difference between May and July was not significant ($p < 0.05$), but it increased significantly in September. This may be because from late spring to summer the temperature in the Baotou area rises, the light is enhanced, and the nutrients (TP and $NH_3-N$) increase, which is conducive to the growth of phytoplankton; however, due to the concentration of rainfall during this period, not only will phytoplankton be diluted, but it will also have a dilution effect on nutrients. The large amount of phytoplankton predation by zooplankton is also one of the reasons for the low phytoplankton biomass in summer. The total biomass increased significantly in September, and the D functional group represented by Bacillariophyta accounted for a large proportion. This is mainly due to the low water temperature and other environmental factors that are more suitable for the growth of Bacillariophyta. The composition of functional groups is similar to that of phytoplankton communities, which are also influenced by environmental factors, and have obvious composition characteristics and successional laws with seasonal changes [28]. The results show that the seasonal variation of dominant functional groups were MP/M/J/D (May), D/F/J/Lo/M/MP/P/TC/S2 (July), and D/MP/Lo/S2 (September). Functional groups D and MP were common advantageous functional groups from May to September, and the BC value was below 0.5, maintaining a relatively stable dominant position.

The representative species of the dominant functional group in spring are *Scenedesmus* sp., *Synedra* sp., *Navicula* sp., and *Microcystis* sp., and their representative habitats are characterized by nutrient−rich turbid shallow water bodies. The increase in spring rainfall in Baotou City and the disturbance of water bodies, coupled with the abundant nitrogen and phosphorus nutrients during this period, promoted the growth and reproduction of these functional groups [29]. With the increase in water temperature and illumination during this period, phytoplankton make full use of the nutrients in water bodies and grow rapidly, and both the number of phytoplankton species and biomass increase significantly.

In summer, the dominant functional groups are more abundant, and the proportions of each functional group are more uniform. Compared with May, the typical indicator species of water experiencing medium eutrophication were added, such as *Kirchneriella* sp. representing the Chlorophyta, and *Fragilaria* sp. of the Bacillariophyta, as well as *Oscillatoria* sp. and *Chroococcus* sp. of the Cyanobacteria. Additionally, the S2 functional group represented by the *Spirulina* sp. of the Cyanobacteria was also added. The habitats indicated are characterized by warm, shallow, highly alkaline water bodies, and because of the high water temperature in summer, they are suitable for the growth of functional group S2, which prefers warm habitats. The biomass was relatively low in July, which was associated with competition among phytoplankton functional groups, ingestion by other aquatic animals, and the heavy consumption of nutrient salts.

In the autumn, the dominant functional groups of the summer also exist, and the functional groups D, MP, Lo, and S2 continue to occupy a dominant position in terms of competition. Compared with summer, the temperature drops in autumn, the duration of daylight becomes shorter, and the dominant functional groups are less abundant than in summer, so the degree of eutrophication in autumn is lower than in summer.

Similar to the evaluation results of the *Q* index and the comprehensive nutrition index, Nanhai Lake as a whole was found to be at a mesotrophic level. Through the Pearson correlation analysis of environmental factors and the *Q* index and *TSIM* index, the two indexes were found to have significant correlation with nutrients ($p < 0.05$). Previous

studies have shown that when the amount of rainfall is large in summer, the inflow of water into the lake may cause the nutrient concentration in the lake to decrease, resulting in a low evaluation result of the *TLI* index of the lake in summer. At this time, the reduction in water eutrophication is not due to the improvement in water quality or the reduction in biomass, but the reflection of physical and chemical factors [8]. This differs from the evaluation results of the *Q* index and *TSIM* index in the lake. Compared with the other two months, the water conditions were poor and the degree of eutrophication was high in July. Some scholars conducted a comprehensive trophic index evaluation of the phytoplankton community structure in new urban landscape lakes. This showed that the lakes were in a mild eutrophic stage, with a poor water source, unstable substrate, and many construction projects ongoing around the lake shore, which affected the transparency and turbidity of the water body, resulting in the poor stability of the phytoplankton community structure [30]. As an important scenic lake in Baotou, Nanhai Lake is inevitably affected by man−made and natural conditions. The water body is turbid and long−term pollutants are deposited, and the water pollution is aggravated. However, the water body has been established for a long time, and it has a certain self−purification ability. Therefore, compared with the newly built landscape lakes, its trophic degree is relatively stable, and the trophic index is relatively low. However, the self−regulation ability of lakes is limited, and sometimes is not enough to deal with the harm caused by pollutants.

### 4.2. The Relationship between Phytoplankton Functional Groups and Environmental Factors

The stratification and convection of water temperature in reservoirs due to seasonal changes is an important feature. For waters with a certain water depth, the temperature difference of water temperature stratification is large, the duration is long, and the phenomenon of seasonal stratification is present, which affects the physical characteristics, chemical reaction processes, and biological activities of the water body, especially the vertical distribution of phytoplankton [31–33]. Fu et al. found that the density of algae cells in a surface water body was significantly higher than that in the middle and lower layers when there was stratification, and the upper and lower water bodies were evenly mixed and layered in winter and disappeared, the density difference of algae in each water layer was small, and the dominant algae were replaced by Bacillariophyta [34]. Li et al. found that the proliferation of phytoplankton in the upper layer of stratified water bodies in the Yangtze River Estuary and its adjacent water is a necessary condition for the formation of hypoxia, and Bacillariophyta have a high sedimentation rate, so the community dominated by Bacillariophyta is more conducive to the formation of low oxygen [35]. This suggests that water stratification affects algae growth and distribution. The water samples taken in this study are mainly mixed water bodies, so the research content on water stratification and phytoplankton communities is insufficient. Subsequently, it is not only necessary to take samples for many months, but also to analyze different water layers as important factors affecting the community structure of phytoplankton, in order to provide more perfect and scientific data for the evaluation of the quality of the study area.

In addition to the phenomenon of water stratification, water temperature and nutrients have a major impact on the growth and distribution of algae, and other physical and chemical factors affect nutrients and water temperature under certain conditions, subsequently affecting the distribution of phytoplankton communities in the water body [36–38]. In this study, through the Spearman correlation analysis and RDA analysis of dominant functional groups and environmental factors, the results show that various nutrients ($Mg^{2+}$, $Na^+$, $K^+$, $Ca^{2+}$, TP, $NH_3-N$, $SO_4^{2-}$, and $NO_3^-$) and COD, WT, SD, and Chl−a are all important factors affecting the succession of phytoplankton functional groups.

Previous studies have shown that the contribution of water depth and transparency to the phytoplankton community structure is mainly reflected in the impact on light and water mixing. With the arrival of the rainy season, the depth of the lake increases, accompanied by the proliferation of thermophilic phytoplankton, and the transparency of the water body decreases [39]. On the other hand, with an increase in transparency, the abundance and

biomass of phytoplankton and some aquatic organisms gradually decreased [38]. In May, the climate of the study area continues to heat up, the illumination increases, and the rainy season arrives, which is suitable for the growth of thermophilic algae, therefore having a certain correlation with transparency. Functional group P and F, as intermediate eutrophic nutrient indicator species, were unique functional groups in July, and had a significant positive correlation with SD ($p < 0.05$).

WT is the most direct factor affecting the succession of phytoplankton communities [40], and has a significant correlation with functional groups D, S2, P, and F ($p < 0.05$). Functional group D is represented by Bacillariophyta that have a suitable water temperature of 18.49~21.12 °C for growth and belong to the group of algae that prefer low temperatures. The habitat indicators of functional groups P, S2, and F, dominated by Chlorophyta, were all suitable for a warm−water environment, so there was a significant positive correlation with water temperature ($p < 0.05$). Studies have shown that within a certain range, organic matter can promote the growth and metabolism of phytoplankton. The large increase in phytoplankton biomass increases water $COD_{Mn}$, so this may be the reason why COD, TC, P, S2, Lo, and other functional groups in this study were significantly positively correlated ($p < 0.05$) [41,42]. Similarly, the increase in Chl−a concentration is also evidence for the increase in phytoplankton biomass [43]. From May to September, the Chl−a concentration gradually increased, and the phytoplankton biomass also increased with time. Therefore, Chl−a has a certain correlation with the growth of various phytoplankton.

Heavy metals are important factors affecting the growth of algae. According to the "GB Surface Water Environmental Quality Standard" (3838−2002), the heavy metal content did not exceed the standard. According to the anion and cation detection results, $Cl^-$, $SO_4^{2-}$, and $Na^+$ were three kinds of salts with a relatively high content, seriously exceeding the standard. This study showed that salt ions were the main factor affecting the distribution of functional groups. Some scholars have studied Hulun Lake [44], a highly alkaline water body in a typical cold area in the northern inland region, and found that nutrients are the most important driving factor for phytoplankton growth in summer. Some scholars studied the saline alkali lakes in Heilongjiang and found that the nutrient content and hydrological conditions were the main reasons for the seasonal succession of functional groups in these lakes [45]. $Na^+$, $Mg^{2+}$, $Ca^{2+}$, and $K^+$ are significantly correlated with various functional groups. Some studies found that $Ca^{2+}$, $Na^+$, $Mg^{2+}$, and $K^+$, as the main cations in water, had a great influence on the growth and existence of *Microcystis* sp. [46]. Too−high or too−low concentrations of cations will inhibit the growth of algae. Previous studies have shown that salt, TDS, TN, TP, $NH_3-N$, and $NO_3-N$ are all nutrients that affect the growth and reproduction of phytoplankton [47].

A survey of Guanting Reservoir found that the representative functional groups Wo/H1/P/D/C/N grew vigorously with the increase in TP and $NH_3-N$ contents in summer and autumn. In autumn, although the water temperature decreased, under the condition of sufficient nutrient salts, the algae of functional groups TC and H1 were still the dominant groups [48]. However, this study found that TP was significantly negatively correlated with functional groups D and Lo. TP reached the standard limits of Class V, Class IV, and Class III surface water in three quarters. With the decrease in TP, functional groups D and Lo became stable dominant species in summer and autumn. $NH_3-N$ exceeded the range of Class V surface water throughout the year, and was significantly positively correlated with functional groups MP, J, and M. From May to September, the $NH_3-N$ content also gradually decreased, but the excess was still serious; this was sufficient for the survival of functional groups such as MP, J, and M, which are suited to turbid and high−nutrient environments. Functional group D was extremely significantly ($p < 0.01$) and significantly ($p < 0.05$) negatively correlated with $SO_4^{2-}$ and $NO_3^-$, respectively. Functional group D was suitable for growth in the $SO_4^{2-}$ concentration range of 311.23~326.60 mg/L and $NO_3^-$ concentration range of 11.90~12.80 mg/L. Previous studies have shown that the D group is mainly dominated by Bacillariophyta, and Bacillariophyta usually like to live in water environments with high nitrate concentrations [41,49,50], indicating that the level of



nitrate content is relatively high. With the continuous increase in nutrient concentration, the biomass of D functional group was restricted to a certain extent.

## 5. Conclusions

In 2021, a total of 7 phyla and 68 genera of phytoplankton were identified and classified into 23 functional ecological groups, and it was found that the dominant functional groups in summer are richer. Functional groups D and MP have long been dominant in the lake, the suitable habitat for which is turbid water bodies. The *Q* value was between 1.94 and 3.13, and the *TSIM* value was between 49.32 and 52.11, indicating that Nanhai Lake is in a mesotrophic state.

Correlation analysis and RDA analysis showed that various nutrients ($K^+$, $Na^+$, $Mg^{2+}$, $NH_3-N$, $HCO_3^-$, $Ca^{2+}$, $SO_4^{2-}$, $Cl^-$, TP, and $NO_3^{2-}$), SD, COD, WT, and Chl–a were the main environmental factors affecting the biomass of dominant functional groups. Nutrients (TP, $NH_3-N$, $Cl^-$, $SO_4^{2-}$, and $Na^+$) and $BOD_5$ exceeded the national surface water quality standards to varying degrees. This showed that salt was a major factor affecting the water quality during the study period. Overall, the eutrophication problem in the lake is relatively serious. It has long been established. Although it has a certain ability to regulate its own pollution, it is still affected by human and natural factors. If this is not treated, there is still the possibility of cyanobacteria outbreaks.

In summary, this study only selected one month in spring, summer, and autumn for sampling, and it only reflects the changes of water and phytoplankton in the Nanhai Lake during a relatively short period of time. Therefore, this is only our preliminary research, and in the next stage it is necessary to conduct continuous investigation of algae community characteristics and water quality factors in different seasons and years for water bodies, and to fully consider the influence of water stratification, hydrodynamic conditions, and other factors on the functional group structure of phytoplankton, so we will continue to carry out experiments and conduct long−term in−depth research to reveal them.

**Author Contributions:** Conceptualization, D.G.; validation, Z.W. and X.J.; statistical analysis and taxonomic determination, Z.G.; investigation, Z.G. and W.W.; writing—original draft preparation, Z.G.; writing—review and editing, Z.G., D.G., and J.B.; funding acquisition, D.G. All authors have read and agreed to the published version of the manuscript.

**Funding:** This study was supported by the National Natural Science Foundation of China (No. 31660151), the Natural Foundation of Inner Mongolia (No. 2020LH03003), the Science and Technology Major Special of Ordos (No. 2021 ZD I 22−6), and the Inner Mongolia Science and Technology Planning Project (2022YFXZ0037).

**Institutional Review Board Statement:** Not applicable.

**Data Availability Statement:** Not applicable.

**Acknowledgments:** The authors would like to thank all those involved in the implementation of this study for their assistance, guidance, and support.

**Conflicts of Interest:** The authors declare no conflict of interest.

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
