# Peer review of "Phytoplankton Community Structure and Its Relationship with Environmental Factors in Nanhai Lake"

_diversity, doi:10.3390/d14110927_

Round 1
Reviewer 1 Report
In the reviewed paper interesting results about phytoplankton community structure of Nanhai Lake are presented. During the study traditional phycological methods together with the estimation of ecological factors were used. The manuscript is important for understanding the ecology of freshwater algae. But before the acceptance it is necessary to improve the quality of the paper.
I recommend the manuscript for publication with major revision.
Remarks to the authors.
First of all, use the Diversity template with the line numbering.
What means “SD, COD, WT, and Chl.a” in Abstract?
For identification different groups of algae it is not enough to use only one key book. Maybe you used some other sources and databases (for example, Algaebase)?
How many species were identified?
In estimation of community structure use the correct phylum names (Ochrophyta instead Diatoms).
Figure 1 should be situated in the center of the page. What means N1-N5? Add the designations.
Table 2: Add authors of genera and species. Add horizontal lines, otherwise, the correspondence between the 2nd and 3rd column is unclear.
Figure 2: Place the image in the center of the page.
Figure 3: What mean H1-H6? Add the explanations.
Figure 4: Place the image in the center of the page.
Figure 5: Axis names – Add the space between “Ecological status index” and “(Q)”, “Comprehensive nutrition status index” and “(TSIM)”.
Figure 6: What mean (a) – (k)?
Figure 7: What mean (a), (b)?
Figure 8, 9: Place the image in the center of the page. Image quality needs to be improved.
Figure 10: What mean (a)-(i)?
Figure 11: Place the image in the center of the page. Image quality needs to be improved. The yellow stars are almost invisible.
In Conclusion instead the listing the functional groups from Y to W1 it is better to say something like “functional ecological groups”. Use clear terms instead of letters.
Author Response
Dear reviewer
I have revised the article according to your suggestions. If there is anything wrong, please inform me. Thank you.
Have a good day.
Sincerely
Donghui

Author Response

(The authors gave the same response as above.)

Reviewer 3 Report
Review for the paper "Phytoplankton Community Structure and Its Relationship with Environmental Factors in Nanhai Lake" by Donghui Gong, Ziqing Guo, Wenxue Wei, Jie Bi, Zhizhong Wang, Xiang Ji submitted to "Diversity".
General comment.
Phytoplankton are key primary producers in any aquatic ecosystem. They represent a community of photosynthetic microorganisms which differ considerably in their sizes. In the freshwater environment, plankton populations may exhibit considerable variability in terms of composition, abundance and community structure. To assess temporal succession in phytoplankton populations, we must have data for different seasons. This study aimed to reveal the seasonal pattern of the phytoplankton in a Chinese lake. The authors also tried to seek factors determining variations in phytoplankton composition and community structure. Although the sampling intervals were not fine enough, this study provides general evidence for seasonal differences in the phytoplankton. The topic of the paper is in the scope of the Journal and the results obtained in the study may be interesting for monitoring of lake ecosystems. Standard methods to collect phytoplankton samples were used in the study. Statistical treatment seems to be valid and the main results are confirmed by relevant criteria. Illustrations are rather good and provide real patterns. However, the current ms has some major flaws (see also comments below). First, the data were not sufficiently statistically analyzed and argued. This further precluded the clear demonstration of the phytoplankton community structure. Second, the methodology is not well described.
Major concerns.
Material and methods. Statistical treatment must be improved. I suggest using cluster/NMDS analysis to investigate the community structure of phytoplankton in the study area.
Material and methods. Also, seasonal comparisons of the main phytoplankton measures (contribution of major taxa and biological indices) must be done using relevant methods such as ANOVA or Kruskal-Wallis test.
Specific remarks.
Abstract. Please, use the full definition for MP, D, SD, COD, and WT instead of the abbreviations. Also, describe more the fictional groups identified in the study.
Introduction. The authors must clearly define the main goal of the study and highlight the novelty of the study.
Material and methods. Environmental background (short description of climatic conditions, general hydrology, and bottom topography) is needed to be included in the ms to provide better presentation of the study area.
Section 2.1. Please, provide data on the sample size for each season (total number of stations, samples, measurements).
Fig. 1. Coordinates must be indicated on the map.
Section 2.2.Mesh size of the plankton net must be indicated.
Section 2.2.Consider replacing "appraisals" with "replicates".
Section 2.2. A procedure to identify phytoplankton taxa, calculate abundance/biomass must be described in detail.
Section 2.2. The authors must give a detailed description of the method to delineate different functional groups of the phytoplankton.
Section 2.3. and Table 2. Explain, what do mean TP, COD, and BOD.
Results. Mean values should be presented with SD or SE.
Section 3.1. and Section 4.1. The authors should correct the Latin names of taxa taking into account that only species and genus names must be in italics while the names of higher taxa must be in ordinary font.
Section 3.2.1. Fig. 4. The seasonal changes in the community turnover values were not demonstrated clearly. Think of a better way of presentation.
Fig. 7 caption. The authors noted in Section 3.2.3 that ‘the content of cations in water from May to September ranged from 441.16 to 543.29 mg/L’. However, the Fig. 7 caption indicates that anion and cation concentrations were in phytoplankton. Please, correct. Also, replace ‘Ion species’ with ‘Ion concentration’ in Fig. 7.
Fig. 8. I would like to see a table with concentrations of heavy metals instead of Figure 7.
Author Response

(The authors gave the same response as above.)

Round 2
Reviewer 1 Report
Dear authors,
You have corrected the MS according my suggestions. I can recommend it for publication in the present form.
Good luck
Author Response
Dear Reviewer
First of all, I would like to express my thanks for your pertinent suggestions for the paper, which is of great help to improve the quality of the paper. And I will revise the paper as suggested by the academic editor. Thanks again.
Have a good day.
Donghui
Reviewer 2 Report
Good job
The article is now complete well and can be accepted for publication after proofreading for spelling errors again.
Good luck
Author Response

(The authors gave the same response as above.)

Reviewer 3 Report
The authors have revised the paper accroding to my suggestions.
Author Response

(The authors gave the same response as above.)
